# Nonlinear Inference Learning for Differentially Private Massive Data

## Abstract

The Bag of Little Bootstraps (BLB) method is widely utilized as a robust and computationally efficient approach in statistical inference studies involving large-scale data. However, this sampling technique overlooks the privacy protection of the original data. To address this limitation, we enhance the existing differential privacy algorithm and integrate it with the BLB method. This integration gives rise to a novel differential privacy mechanism, enabling a comprehensive statistical analysis of aggregated parameters while safeguarding the confidentiality of individual private data. Additionally, to address both the variability in noise variance under the differential privacy mechanism and the uncertainty surrounding estimate distributions, we employ the central limit theorem within the context of nonlinear expectation theory. This facilitates the derivation of the corresponding test statistic and the introduction of a hypothesis testing methodology. Furthermore, we validate the commendable performance of our proposed inference procedure through data simulation studies. The big data-oriented differential privacy-preserving mechanism proposed in this study effectively fulfills the requirements for privacy preservation without compromising subsequent statistical inference. This contribution holds significant reference value for the sharing of pertinent data and endeavors related to statistical analysis.

## 1 Introduction

In the era of massive data, the expansion of dataset sizes is surging, and large-scale datasets are becoming more prevalent. Simultaneously, as technology for processing massive data matures and computer hardware performance improves, using massive data becomes more efficient. However, significant sample sizes, inherent variability in sampled data, and uncertainty around model selection collectively pose new challenges to statistical modeling. Additionally, ensuring user privacy remains crucial. Hence, our research focuses on preserving data privacy while conducting effective statistical analyses.

Bootstrap methods introduced by Efron (1992) and subsequent advancements, like subsampling Politis et al. (1999) and out-of-bootstrap methods Bickel & Sakov (2008), have enhanced statistical inference for large-scale data sampling. Despite optimal resample size strategies Bickel et al. (2012), substantial computation remains necessary. The Bag of Little Bootstraps (BLB) method by Kleiner et al. (2012) combines outcomes from small bootstrapped subsets of a larger dataset, benefiting statistical inference in extensive datasets. Merging bootstrap statistics with computation through simulation yields robust results and enhanced efficiency.

However, prior studies concentrating on properties of estimators from extensive data methods overlook sampled data privacy. If data providers distrust collectors or analysts, obtaining accurate data is challenging, hindering inference. Differential Privacy (DP), proposed by Dwork et al. (2006), is a privacy definition based on noise injection. Noise can be added to data, model parameters, or outputs, preventing privacy attacks. Noise-based privacy methods are efficient but can reduce accuracy.

There is extensive literature on massive data with differential privacy (Sarwate & Chaudhuri, 2013; Joseph et al., 2018; Li et al., 2021). Xiong et al. (2018) developed an efficient, differential private frequent itemsets mining algorithm over large-scale data. Vasa & Thakkar (2022) presented different methods for protecting privacy for deep learning in massive data analysis. The methods reviewed in

Vasa & Thakkar (2022) are classified based on anonymization, optimization-based approaches, and cryptographic methods. Recent work attempt to link differential privacy to statistical problems of streaming data by developing privacy-preserving algorithms. For example, Duchi et al. (2014) proposed a common local Differential Privacy (LDP) for streaming data with SGD algorithm to obtain a valid point estimation. Cummings et al. (2018) constructed a novel dynamic setting to provide an accurate analysis of empirical risk minimization under the DP mechanism for a steaming database. Hasidim et al. (2020) established a connection between the adversarial robustness of streaming algorithms and the DP mechanism. Although there have been many advances in differential privacy since the pioneering work of Dwork et al. (2006), one could argue that due to the lack of extensive statistical inference guidelines, their practical utility in massive data area is minimal. In particular, there is no general procedure for performing statistical hypothesis testing in the context of massive data, which our paper plans to overcome.

This paper introduces an algorithm for implementing a differential privacy mechanism for sample statistics, utilizing the Bag of Little Bootstraps sampling method. The overall estimate is derived through an average-based aggregation technique applied to privacy-treated sample statistics. However, the variance of the noise introduced by the differential privacy algorithm exhibits heterogeneity due to varying sensitivities among the bag-specific samples. This variance disparity results in uncertainty regarding the distribution of the analyzed statistics, consequently impeding effective statistical inference. To address this, we employ the sublinear expectation theory advanced by Peng (2010), enabling an exploration of asymptotic statistical outcomes. Specifically, we establish the asymptotic G-normal distribution of terminal DP estimates, enabling valid statistical inferences about overall parameters and assessment of statistic performance via numerical simulations.

The primary contributions of this paper are outlined below:

- Pioneering differential privacy in static massive data using a novel algorithm.

- Providing a general, scalable differentially private algorithm adaptable to various methods.

- Analyzing asymptotic distribution of statistics using Nonlinear expectation theory, constructing parameter statistics and hypothesis testing.

The paper's structure is outlined as follows. We commence by introducing our approach for formulating differentially private estimators within the scope of massive data. Subsequently, we delve into an exploration of their asymptotic characteristics, elucidating the methodology for creating corresponding hypothesis tests and confidence intervals. Due to the length of the article, please refer to the appendix for more details on this section. We substantiate the effectiveness of inference using our differential privacy mechanism through the examination of synthetic data. The final section encapsulates our article by offering conclusions, a comprehensive discussion of outcomes, and insights into potential future research avenues.

## 2 METHODOLOGY

### 2.1 BLB METHOD

The original dataset is represented as $\mathbb{X} = [X_1, X_2, \cdots, X_N]$, encompassing a substantial number of $N$ data points. We subsequently engage in $K$ non-replacement sampling iterations on the initial dataset to procure $K$ mutually independent sub-samples, denoted as $\mathbb{X}^* = \{\boldsymbol{X}_1^*, \boldsymbol{X}_2^*, \cdots, \boldsymbol{X}_K^*\}$. Within this context, for $1 \leq k \leq K$, each sub-sample $\boldsymbol{X}_k^* = \left[X_{k,1}^*, X_{k,2}^*, \cdots, X_{k,r_k}^*\right]$ contains a total of $r_k$ data points.

From each sampled dataset, we conduct bootstrap sampling with replacement, yielding $B$ bootstrap samples. For instance, with the subsample $\boldsymbol{X}_k^*$, we form bootstrapped samples $\left\{\boldsymbol{X}_k^{*1}, \boldsymbol{X}_k^{*2}, \cdots, \boldsymbol{X}_k^{*B}\right\}$. For $1 \leq b \leq B$, each bootstrapped sample $\boldsymbol{X}_k^{*b}$ comprises $r_k$ data points, allowing us to derive an estimate $\widehat{\theta}_k^{*b}$ using $\boldsymbol{X}_k^{*b}$. The collection $\left\{\widehat{\theta}_k^{*b}\right\}_{b=1}^{B}$ is then aggregated to derive a corresponding estimate $\widehat{\theta}\left(\boldsymbol{X}_k^*\right) = \widehat{\theta}_k \in \Theta$ for the unknown population parameter $\theta$ associated with $F$. In this manner, we obtain $K$ estimates of the parameter $\theta$. Moreover, we designate $Q_N(F)$

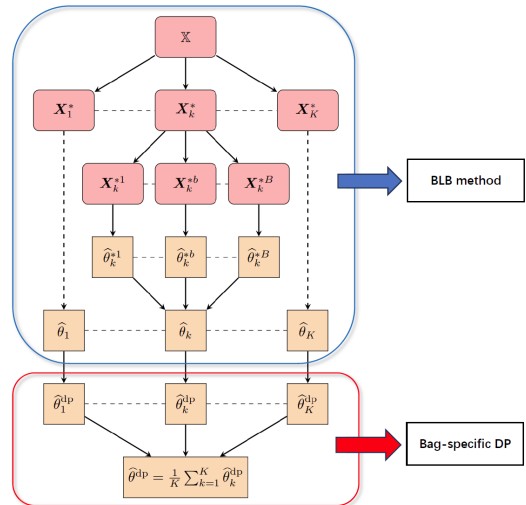

Figure 1: Flowchart of BLB and Bag-specific DP method

as the authentic underlying distribution of $\theta$, a determination influenced jointly by the form of $F$ and the parameter $\theta$.

The classical goal of the BLB is to compute an estimator quality assessment $\psi\left(Q_N(F), F\right)$, where $\psi$ can denote quantile, confidence interval, standard error, or deviation. For example, if $\psi$ computes a confidence interval, it might derive the distribution of the statistic $\sqrt{N}(\widehat{\theta} - \theta)$. In practice, since $F$ or $Q_N(F)$ is unknown, we must estimate $\psi\left(Q_N(F), F\right)$ based on the subsample-based estimators.

## 2.2 BAG-SPECIFIC DIFFERENTIAL PRIVACY

Considering privacy protection, this approach becomes inadequate for the diverse range of applications involving continuous subsampling over time without privacy safeguards. This limitation arises from the tendency of subsampled data sets to contain bag-specific sensitive information, where each bag corresponds to only one subsampled data set. Despite the trust data providers place in data collectors or managers, doubt may arise concerning data learners employing machine learning techniques, especially when their statistical algorithms identify sensitive observations or outliers. To address privacy concerns, we propose the incorporation of a differential privacy mechanism into the bag-specific estimator $\widehat{\theta}_k$. Firstly, we introduce the notions of bag-specific sensitivity and $(\varepsilon, \delta)$-differential privacy for this purpose.

**Definition 1** (**Bag-specific sensitivity**) *The $\ell_1$-sensitivity of bag-specific estimate $\widehat{\theta}_k$ is defined by*

$$\Lambda_k = \sup_{\left\{\delta\left(\boldsymbol{X}_k^*, \boldsymbol{X}_{k'}^*\right)=1\right\}} \left|\widehat{\theta}\left(\boldsymbol{X}_k^*\right) - \widehat{\theta}\left(\boldsymbol{X}_{k'}^*\right)\right| < \infty$$

*where $\delta\left(\boldsymbol{X}_k^*, \boldsymbol{X}_{k'}^*\right)$ represent the Hamming distance between two neighboring dataset $\boldsymbol{X}_k^*$ and $\boldsymbol{X}_{k'}^*$. That is, we can use $\Lambda_b$ to calculate the sensitivity of $\widehat{\theta}_k$ uniquely determined by the bag-specific sample $\boldsymbol{X}_k^*$.*

**Definition 2** ($(\varepsilon, \delta)$-**differential privacy**) *For an estimator $\widehat{\theta}_k$, satisfying $(\varepsilon, \delta)$-differential privacy, if for any two adjacent data sets $\boldsymbol{X}_k^*, \boldsymbol{X}_{k'}^*$, for $F \subseteq \mathrm{Range}(\widehat{\theta})$, the estimator obeys the following inequality*

$$\Pr\left[\widehat{\theta}\left(\boldsymbol{X}_k^*\right) \in F\right] \le e^{\varepsilon_k} \cdot \Pr\left[\widehat{\theta}\left(\boldsymbol{X}_{k'}^*\right) \in F\right] + \delta_k$$

where the values of $\varepsilon_b$ and $\delta_k$ are dictated by the bag-specific privacy requisites pertaining to the $b$-th bag's samples. When the privacy stipulations are uniform across distinct subsamples, we can employ a unified set of values, denoted as $\varepsilon_k = \varepsilon$ and $\delta_k = \delta$. For the sake of generality, this article assumes uniform privacy requirements across all bags.

In this study, we incorporate the $(\varepsilon, \delta)$-Gaussian differential privacy mechanism into the estimator $\theta_k$ by introducing random noise $u_k$ drawn from a Gaussian distribution $N\left[0, \sigma_k^2\right]$. This article's approach ensures the same privacy standards for all bags. The subsequent lemma asserts that the Gaussian mechanism applied to each sample bag guarantees $(\varepsilon, \delta)$-differential privacy.

**Lemma 1** *Dwork et al. (2014) Let $\varepsilon \in (0, 1)$ be arbitrary constant. For $c^2 > 2\ln(1.25/\delta)$, the Gaussian Mechanism with parameter $\sigma_k \geq c\Lambda_k/\varepsilon$ is $(\varepsilon, \delta)$-differentially private, where $\Lambda_k$ is the sensitivity of $\widehat{\theta}\left(\boldsymbol{X}_k^*\right)$, for $1 \leq k \leq K$.*

From this lemma, the standard deviation of the added noise is calculated as

$$\sigma_k = \frac{\Lambda_k \sqrt{2\ln(1.25/\delta)}}{\varepsilon}$$

consequently, after privacy processing, the estimator is obtained by $\widehat{\theta}_k^{\text{naive,dp}} = \widehat{\theta}_k + u_k$, and $u_k \sim N\left[0, \sigma_k^2\right]$.

The outlined procedure is illustrated in Fig 1. We performed parameter estimation based on the BLB method and added the bag-specific DP. Our approach entails parameter estimation grounded in the BLB method, augmented with bag-specific differential privacy (DP). Subsequent sections will delve into the statistical inference challenges arising from the introduction of bag-specific DP. However, for $1 \leq k \leq K$, since sub-sampling produces duplication of samples among $\left\{\boldsymbol{X}_k^{*1}, \boldsymbol{X}_k^{*2}, \cdots, \boldsymbol{X}_k^{*B}\right\}$, the individual estimators $\left\{\widehat{\theta}_k^{*1}, \widehat{\theta}_k^{*2}, \cdots, \widehat{\theta}_k^{*B}\right\}$ aren't independent of each other. Additionally, the variance of the distinct estimates' introduced random noise showcases heterogeneity due to varying sensitivities, thereby complicating access to subsequent statistical inference.

## 2.3 DIFFERENTIALLY PRIVATE ALGORITHM

Therefore, we introduce an innovative privacy-preserving algorithm aimed at minimizing the variance fluctuations in random noise. This novel approach facilitates the establishment of hypothesis tests and confidence intervals through the G-Normal theory proposed by Peng & Zhou (2020), thereby enhancing our capacity to address subsequent statistical inference challenges. The framework for generating the novel differentially private online estimator unfolds as follows: Initially, we represent the range of standard deviation fluctuations in random noise as

$$\Delta = \max\left\{\sigma_1, \sigma_2, \cdots, \sigma_K\right\} - \min\left\{\sigma_1, \sigma_2, \cdots, \sigma_K\right\}$$

where $\sigma_1, \sigma_2, \cdots, \sigma_K$ are calculated from the above mentioned Gaussian differential privacy formula.

Next, we specify its upper and lower standard deviation bounds as

$$\bar{\sigma} = \frac{\bar{\gamma}}{\sqrt{n_{\bar{k}}}} + \Delta, \bar{\gamma} = \max_{1 \leq k \leq K}\left\{\sqrt{r_k}\sigma_k\right\}, \bar{k} = \operatorname*{argmax}_{1 \leq k \leq K}\left\{\sqrt{r_k}\sigma_k\right\}$$

$$\underline{\sigma} = \frac{\underline{\gamma}}{\sqrt{r_{\underline{k}}}} + \Delta, \underline{\gamma} = \min_{1 \leq k \leq K}\left\{\sqrt{r_k}\sigma_k\right\}, \underline{k} = \operatorname*{argmin}_{1 \leq k \leq K}\left\{\sqrt{r_k}\sigma_k\right\}$$

Thus our newly constructed differentially private estimator is $\widehat{\theta}_k^{\text{dp}} = \widehat{\theta}_k + \omega_k = \widehat{\theta}_k + \xi_k \epsilon_k$, where $\epsilon_k$ is a random variable that obeys a normal distribution $N[0, 1]$, $\{\xi_k\}_{1 \leq k \leq K} \in \Sigma(\underline{\sigma}, \bar{\sigma})$ denotes the collection of all measurable sequences $\{\mathcal{F}_k\}_{k=1}^{\infty}$, which bounded to $[\underline{\sigma}, \bar{\sigma}]$. On the other hand, the natural filtration is generated by $\omega_k$ and the value of $\xi_k$ is taken as

$$\xi_k = \begin{cases} \bar{\sigma}, & |S_{k-1}|/\sqrt{K} \leq \lambda_{k-1}^2 c_{1-\alpha/2} \\ \underline{\sigma}, & |S_{k-1}|/\sqrt{K} > \lambda_{k-1}^2 c_{1-\alpha/2} \end{cases}$$

where $S_k$ equals to

$$\sqrt{r_1}\left(\widehat{\theta}_1^{\mathrm{dp}} - \frac{1}{k}\sum_{i=1}^k \widehat{\theta}_k\right) + \cdots + \sqrt{r_k}\left(\widehat{\theta}_k^{\mathrm{dp}} - \frac{1}{k}\sum_{i=1}^k \widehat{\theta}_k\right)$$

$c_{1-\alpha/2}$ is critical value of $t$ distribution with proper freedom and it can be approximated by $\Phi^{-1}(1 - \alpha/2)$ when $K$ tends to infinity, $\lambda_k^2 = \mathrm{var}\left[\sqrt{r_k}\widehat{\theta}_k^{\mathrm{dp}}\right]$ is calculated as

$$\lambda_k^2 = \frac{1}{k}\sum_{i=1}^k \left(\sqrt{r_k}\widehat{\theta}_k^{\mathrm{dp}}\right)^2 - \left\{\frac{1}{k}\sum_{i=1}^k \sqrt{r_k}\widehat{\theta}_k^{\mathrm{dp}}\right\}^2$$

therefore, we end up this section with a new differentially private estimator $\widehat{\theta}_k^{\mathrm{dp}}$. Moreover, it can be verified that this new estimator still satisfies Gaussian differential privacy requirement, i.e., Theorem 1 .

**Theorem 1** *The bag-specific estimator $\widehat{\theta}_k$ satisfies $(\varepsilon, \delta)$-Gaussian differential privacy if the standard deviation is produced by above policy for $k = 1, \cdots, K$.*

Theorem 1 holds, because the standard deviation of the noise we defined satisfies the following

$$\bar{\sigma} = \frac{\bar{\gamma}}{\sqrt{n_{\bar{k}}}} + \Delta > \sigma_k = \frac{\Lambda_k\sqrt{2\ln(1.25/\delta)}}{\varepsilon}$$

$$\underline{\sigma} = \frac{\underline{\gamma}}{\sqrt{r_{\underline{k}}}} + \Delta > \sigma_k = \frac{\Lambda_k\sqrt{2\ln(1.25/\delta)}}{\varepsilon}$$

---

**Algorithm 1** Differentially Private Algorithm

---

**Input:** $K, \widehat{\theta}_k, \Lambda_k, \epsilon_k$
**Function:** Differential Privacy Estimator
    **for** $k = 1$ to $K$ **do**
        Compute standard deviation of Gaussian DP noise $\sigma_k$
        Compute statistics $S_k$
        Compute asymptotic variance $\lambda_k^2$
        Generate standard normal distribution variable $\epsilon_k$
    **end for**
    Compute fluctuation range $\Delta$
    Compute upper and lower bounds $\bar{\sigma}, \underline{\sigma}$
    **for** $k = 1$ to $K$ **do**
        **if** $|S_{k-1}|/\sqrt{K} > \lambda_{k-1}^2 c_{1-\alpha/2}$ **then**
            $\xi_k = \bar{\sigma}$
        **else**
            $\xi_k = \underline{\sigma}$
        **end if**
    **end for**
    **return** $\widehat{\theta}_k^{\mathrm{dp}} = \widehat{\theta}_k + \xi_k\epsilon_k$
**End Function**

---

We delve into the asymptotic properties of the estimators $\left\{\widehat{\theta}_k^{\mathrm{dp}}\right\}_{k=1}^K$. For details, please refer to the Appendix.

## 3 HYPOTHESIS TEST AND CONFIDENCE INTERVAL

### 3.1 TAIL CAPACITY

The parameter $p$, referred to as the tail capacity of the G-Normal distribution, is a term in which "capacity" serves as a broader concept akin to probability. The determination of $p$ holds essential

significance for comprehending the G-Normal distribution, and it is imperative to pinpoint the value of $p$ corresponding to a conventional significance level such as $\alpha = 0.05$. Initially, we delve into the examination of the right tail capacity as follows

$$p_1 \left\{ \bar{\lambda}\Phi^{-1}(1-\alpha); \underline{\lambda}, \bar{\lambda} \right\}$$

$$= \lim_{K \to \infty} \sup_{\{\lambda_k\} \in \Sigma(\underline{\lambda}, \bar{\lambda})} \Pr \left\{ \sum_{k=1}^{K} \frac{\sqrt{r_k}}{\sqrt{K}} \left( \widehat{\theta}_k^{\mathrm{dp}} - \theta \right) > \bar{\lambda}\Phi^{-1}(1-\alpha) \right\}$$

besides, we can obtain the following Theorem

**Theorem 2** *for any $\alpha \leq 0.05$ and $\underline{\lambda} < \bar{\lambda}$, the right tail capacity is*

$$p_1 = p_1 \left( \bar{\lambda}\Phi^{-1}(1-\alpha); \underline{\lambda}, \bar{\lambda} \right) = \frac{2\alpha}{1 + \underline{\lambda}/\bar{\lambda}}$$

It's important to note that when the estimators $\left\{ \widehat{\theta}_k^{\mathrm{dp}} \right\}_{k=1}^{K}$ are independently generated within each bag, indicating the absence of duplicate samples across subsamples and the application of a common Gaussian mechanism for perturbation (i.e., $\underline{\lambda} = \bar{\lambda}$), the right tail capacity maintains asymptotic equivalence to the classical significance level $\alpha$ as $K \to \infty$. However, due to non-identity originating from the sequence of heterogeneous variances $\{\lambda_k\}_{k=1}^{\infty}$, the assumption of identical behavior may be compromised. This leads to an increment in the right tail capacity by a minimum of $2\bar{\lambda}/(\underline{\lambda} + \bar{\lambda})$. The definition of the right tail capacity in Theorem 3 suggests that selecting the critical value of the classical standard normal distribution $\Phi^{-1}(1-\alpha)$ results in a right tail capacity of $\frac{\alpha}{1+\underline{\lambda}/\bar{\lambda}}$. Thus, we can approximate the two-tailed capacity using the original critical value

$$p_2 \left\{ \bar{\lambda}\Phi^{-1}(1-\alpha/2); \underline{\lambda}, \bar{\lambda} \right\} \approx 2p_1 \left\{ \bar{\lambda}\Phi^{-1}(1-\alpha/2); \underline{\lambda}, \bar{\lambda} \right\}$$

$$= \frac{2\alpha}{1 + \underline{\lambda}/\bar{\lambda}}$$

## 3.2 TEST STATISTIC

Next, we proceed to construct classical test statistics and delve into research concerning statistical inference and hypothesis testing

$$T_K(\theta) = \frac{1}{\sqrt{K}} \sum_{k=1}^{K} \frac{\sqrt{r_k} \left( \widehat{\theta}_k^{\mathrm{dp}} - \theta \right)}{\widehat{\lambda}} \tag{1}$$

where

$$\widehat{\lambda}^2 = \frac{1}{K} \sum_{k=1}^{K} \left( \sqrt{r_k}\widehat{\theta}_k^{\mathrm{dp}} \right)^2 - \left\{ \frac{1}{K} \sum_{k=1}^{K} \sqrt{r_k}\widehat{\theta}_k^{\mathrm{dp}} \right\}^2$$

The null and alternative assumptions are

$$\mathrm{H}_0 : \theta = \theta_0; \quad \mathrm{H}_1 : \theta \neq \theta_0.$$

Given that the variances of the released sequential DP estimators $\left\{ \sqrt{r_1}\widehat{\theta}_1^{\mathrm{dp}}, \cdots, \sqrt{r_K}\widehat{\theta}_K^{\mathrm{dp}} \right\}$ are less than or equal to $\bar{\lambda}^2$, it follows that $\widehat{\lambda}^2$ is also less than or equal to $\bar{\lambda}^2$ with probability one. As a result, if $\theta_0$ represents the true parameter value, we proceed with a test involving the rejection region $\left\{ |T_K(\theta_0)| > \Phi^{-1}(1-\alpha/2) \right\}$. The probability of rejecting the null hypothesis H0 is then bounded from below by $\Pr \left( |S_K(\theta_0)| > \bar{\lambda}\Phi^{-1}(1-\alpha/2) \right)$. However, in cases where the learner has the freedom to select any $\{\lambda_k\} \in \Sigma(\underline{\lambda}, \lambda)$, then

$$\lim_{K \to \infty} \sup_{\{\lambda_k\} \in \Sigma(\underline{\lambda}, \bar{\lambda})} \Pr \left( |S_K(\theta_0)| > \bar{\lambda}\Phi^{-1}(1-\alpha/2) \right)$$

$$= p_2 \left( \bar{\lambda}\Phi^{-1}(1-\alpha/2); \underline{\lambda}, \bar{\lambda} \right)$$

By Theorem 3, the left and right tail capacities are not symmetric in this G-Normal distribution. We can approximate the two tailed capacity by

$$p_2\left\{\bar{\lambda}\Phi^{-1}(1-\alpha/2);\underline{\lambda},\bar{\lambda}\right\} \approx \frac{2\alpha}{1+\underline{\lambda}/\bar{\lambda}} > \alpha$$

Given the non-independence and non-identical nature of the sequential DP estimator sequences $\left\{\widehat{\theta}_k^{\mathrm{dp}}\right\}_{k=1}^{K}$, the type I error rate in this test deviates from $\alpha$. The learner has the capacity to amplify the probability of erroneously rejecting the null hypothesis by a minimum of $2\bar{\lambda}/(\underline{\lambda}+\bar{\lambda})$. As a consequence, if our intention is to uphold the null hypothesis rejection standard at a significant level of $\alpha = 0.05$, the corresponding critical value must be adjusted to

$$\Phi^{-1}\left(1 - \alpha\frac{\underline{\lambda}+\bar{\lambda}}{4\bar{\lambda}}\right)$$

Meanwhile, the $1 - \alpha = 95\%$ confidence interval for $\theta$ till $K$ can be constructed by $[Z_1, Z_2]$

$$Z_1 = \sum_{k=1}^{K}\frac{\frac{\sqrt{r_k}}{\widehat{\lambda}}}{\sum_{k=1}^{K}\frac{\sqrt{r_k}}{\widehat{\lambda}}}\widehat{\theta}_k^{\mathrm{dp}} - \frac{\sqrt{K}\Phi^{-1}\left(1-\alpha\frac{\underline{\lambda}+\bar{\lambda}}{4\bar{\lambda}}\right)}{\sum_{k=1}^{K}\frac{\sqrt{r_k}}{\widehat{\lambda}}}$$

$$Z_2 = \sum_{k=1}^{K}\frac{\frac{\sqrt{r_k}}{\widehat{\lambda}}}{\sum_{k=1}^{K}\frac{\sqrt{r_k}}{\widehat{\lambda}}}\widehat{\theta}_k^{\mathrm{dp}} + \frac{\sqrt{K}\Phi^{-1}\left(1-\alpha\frac{\underline{\lambda}+\bar{\lambda}}{4\bar{\lambda}}\right)}{\sum_{k=1}^{K}\frac{\sqrt{r_k}}{\widehat{\lambda}}}$$

This indicates that the primary cost associated with constructing confidence intervals due to sequential DP under heterogeneous noise is the augmented length of the confidence interval, as $\Phi^{-1}\left(1-\alpha\frac{\underline{\lambda}+\bar{\lambda}}{4\bar{\lambda}}\right)$ is larger than $\Phi^{-1}(1-\alpha/2)$. In reality, $\bar{\lambda}$ and $\underline{\lambda}$ are unknown. We can estimate $\bar{\lambda}^2$ and $\underline{\lambda}^2$ consistently by

$$\widehat{\bar{\lambda}}^2 = \frac{1}{|\bar{I}|}\sum_{k\in\bar{I}}\left(\sqrt{r_k}\widehat{\theta}_k^{\mathrm{dp}}\right)^2 - \left\{\frac{1}{|\bar{I}|}\sum_{k\in\bar{I}}\sqrt{r_k}\widehat{\theta}_k^{\mathrm{dp}}\right\}^2$$

$$\widehat{\underline{\lambda}}^2 = \frac{1}{|\underline{I}|}\sum_{k\in\underline{I}}\left(\sqrt{r_k}\widehat{\theta}_k^{\mathrm{dp}}\right)^2 - \left\{\frac{1}{|\underline{I}|}\sum_{k\in\underline{I}}\sqrt{r_k}\widehat{\theta}_k^{\mathrm{dp}}\right\}^2$$

Let $I$ be the corresponding index multisets where $\bar{I} = \{b : \xi_k = \bar{\sigma}\}$, $\underline{I} = \{k : \xi_k = \underline{\sigma}\}$.

## 4 SIMULATION STUDY

This section scrutinizes the performance of the proposed DP procedure within the context of the mean model. Our assumptions are rooted in the normal distribution $N[2,1]$ characterizing the original data, with the population parameter of interest denoted as $\theta = 2$. Utilizing bag bootstrap sampling, we generate $K$ subsample sets $\{\boldsymbol{X}_1^*, \boldsymbol{X}_2^*, \cdots, \boldsymbol{X}_K^*\}$, stipulating that the number of samples contained in each subsample set ranges from 50 to 100, i.e., $50 \le r_k \le 100, \forall k \in [1, K]$. The mean statistic within the $k$-th bag is expressed as follows

$$\widehat{\theta}_k = \frac{\sum_{i=1}^{r_k}X_{k,i}^*}{r_k}$$

as per the aforementioned equation, when there's a unit change in $X_{k,i}^*$, $\widehat{\theta}_k$ experiences a change of $1/r_k$ units. Hence, the corresponding sensitivity within the $k$-th bag amounts to $1/r_k$. Our approach employs the Gaussian mechanism and the perturbed standard deviation for the naive differential privacy (DP) is given by

$$\sigma_k = \frac{\sqrt{2\ln(1.25/\delta)}}{r_k\varepsilon}$$

Since $r_k$ fluctuates within the range $[50, 100]$, it corresponds to the maximum value of $\sigma_k$ when $r_k = 50$ and the minimum value of $\sigma_k$ when $r_k = 100$. The upper and lower bounds of the standard deviation are thus substituted into the formula

$$\bar{\sigma} = \frac{2\sqrt{2\ln(1.25/\delta)}}{50\varepsilon} - \frac{\sqrt{2\ln(1.25/\delta)}}{100\varepsilon}$$

$$\underline{\sigma} = \frac{\sqrt{2\ln(1.25/\delta)}}{50\varepsilon}$$

Further we can obtain the estimators $\widehat{\theta}_k^{\mathrm{dp}}$ after the differential privacy algorithm processing.

To commence, we analyze the distinction between the asymptotic distribution of $T_K(\theta)$ and the conventional $t$ distribution for the parameters $50 \le r_k \le 100, K = 1000, \delta = 0.01, \varepsilon = 0.1$. In this investigation, we generate a total of 1000 subsamples. Figure 3 illustrates the histogram portraying the empirical distribution of the test statistics $T_K(\theta)$ centered around the veritable value of $\theta$. Upon juxtaposing the histogram against the black density line depicting the classical t distribution with (K-1) degrees of freedom, a noticeable augmentation in the two-tailed capacity becomes evident. This aligns precisely with the theoretical outcomes outlined in Theorem 3.

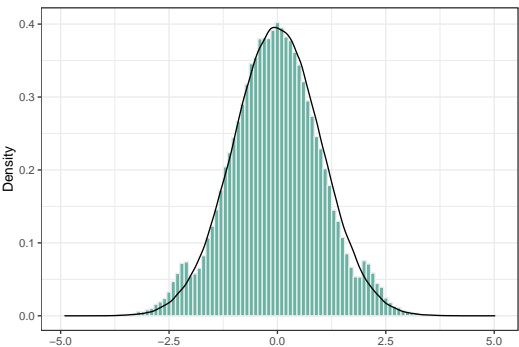

Figure 2: Histogram of the empirical distribution of the test statistics $T_K(\theta)$ under the true value $\theta$. The black line represents the density function of the $t$-distribution with $K - 1$ degrees of freedom.

Subsequently, we delve into the examination of our estimators' confidence intervals. These intervals constitute estimations derived from sample statistics and encase the overall parameters. The evaluation criteria for the adequacy of a confidence interval rest on two principles: firstly, a higher confidence level signifies superior performance; secondly, a narrower interval width implies a more favorable outcome.

Broadly speaking, three primary factors govern the width of a confidence interval. The foremost element is the dispersion exhibited by the complete dataset, typically assessed through variance measurement. Subsequent is the sample size; larger samples yield richer information content, which generally translates into narrower confidence intervals. Moreover, the confidence level itself also influences the interval width.

We purposefully chose $\delta$ values of 0.001, 0.005, and 0.01 to facilitate a comparative analysis of confidence interval widths between estimates processed through the differential privacy mechanism and the original estimates. Figures 3, 4, and 5 offer insights into the $1 - \alpha = 95\%$ confidence intervals of $\theta$, derived from the average-aggregated DP estimators within each bag. Notably, these visualizations demonstrate the congruence between the confidence bands of the estimators based on the original datasets and those produced through larger bag sizes. Concurrently, a diminutive privacy budget combined with a substantial $\delta$ value yields a markedly narrower confidence band, progressively approximating the outcomes obtained through accumulated statistics devoid of DP processing.

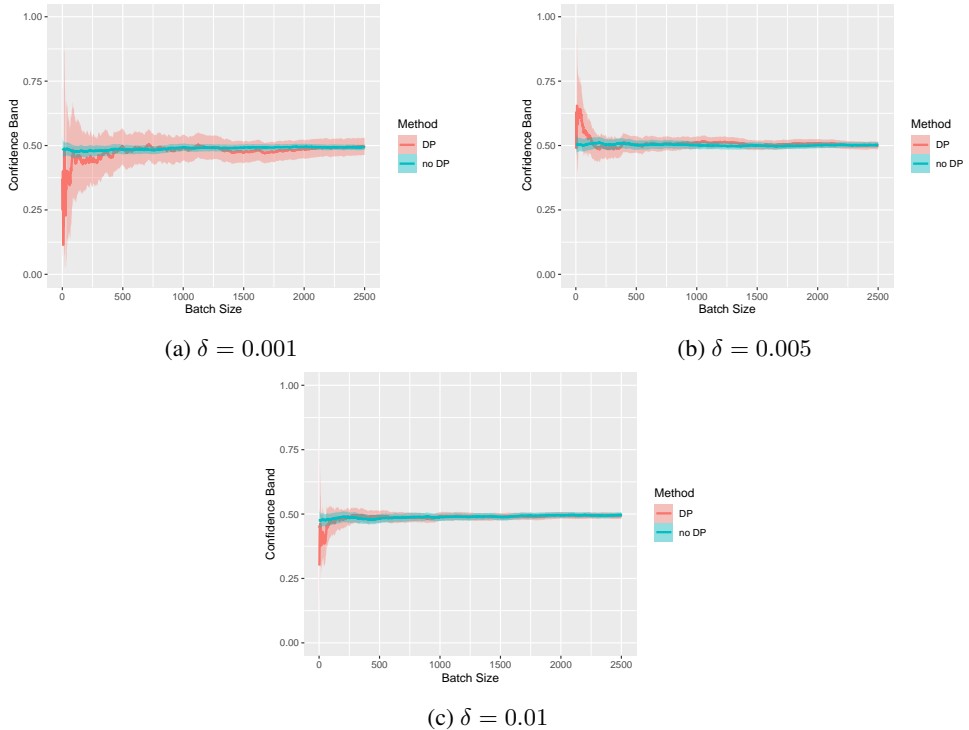

(a) $\delta = 0.001$             (b) $\delta = 0.005$

(c) $\delta = 0.01$

Figure 3: Confidence band of two methods including sequential DP estimation (red) and estimators under original data (green) across three cases $\delta = 0.005, 0.01, 0.01$.

## 5  DISCUSSION

This paper contributes in the following ways: Firstly, we introduce a novel differential privacy estimation method by adapting an existing differential privacy mechanism. This approach enables us to derive estimates for overall parameters of interest through bag bootstrap sampling while safeguarding data privacy by adjusting the $(\varepsilon, \delta)$ parameter within differential privacy. Secondly, we elucidate the process of generating specific estimates for diverse sampled datasets and subsequently obtaining overall parameter estimates. Addressing the intricate realm of statistical inference, we confront challenges arising from the variance uncertainty introduced by privacy protection. To tackle this, we employ the central limit theorem within nonlinear expectation theory. This allows us to establish that the statistics we construct exhibit an asymptotic G-normal distribution as the sample size approaches infinity, yielding favorable statistical properties. In particular, our investigation reveals that the proposed bag-specific sensitivity introduces heterogeneous noisy perturbations with uncertain variances, leading to an enlarged tail capacity for hypothesis and lengthened confidence intervals. Furthermore, potential avenues for improvement and future research are identified. These include extending the sequential $\ell_1$-sensitivity to $\ell_p$-sensitivity and considering statistical inference with local differential privacy, accounting for user-specific privacy settings. Additionally, we propose exploring distributed learning, online learning, or federated learning scenarios, building upon the foundational work of Wasserman & Zhou (2010) and Duchi et al. (2014).

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

# A   APPENDIX

# B   ASYMPTOTICALLY THEORETICAL RESULTS

In this section, we delve into the asymptotic properties of the estimators $\left\{\widehat{\theta}_k^{\mathrm{dp}}\right\}_{k=1}^K$. Our primary concern revolves around assessing whether $\widehat{\theta}_k^{\mathrm{dp}}$ approaches the true value in a meaningful manner as the sample size expands substantially. Specifically, our analysis centers on its asymptotic normality, scrutinizing whether the distribution of the random variable $Z_n = \sqrt{r_k}\left(\widehat{\theta}_k^{\mathrm{dp}} - \theta\right)$ adheres to the normal distribution. If $\widehat{\theta}_k^{\mathrm{dp}}$ functions as an asymptotic normal estimator of $\theta$, it inherently qualifies as a consistent estimator for $\theta$. This coherence arises from the concept of collinearity, pondering whether $\widehat{\theta}_k^{\mathrm{dp}} \to \theta$ holds as $K \to \infty$, here $\to$ signifies the convergence of the random variable within a probability context.

As stipulated by Slutsky's theorem, when an estimator exhibits asymptotic normality, it necessitates convergence in probability, thus ensuring consistent estimation. Within the framework of this paper, our objective entails scrutinizing whether the distribution of the estimator approaches normality as the sampling number K approaches infinity. The mean value of the random variable $Z_n$ equates to $\sqrt{r_k}\left(\widehat{\theta}_k^{\mathrm{dp}} - \theta\right)/K$. Employing the central limit theorem as our foundation, we devise the subsequent test statistic to explore the estimator's asymptotic normality

$$S_K(\theta) = \frac{1}{\sqrt{K}} \sum_{k=1}^K \sqrt{r_k}\left(\widehat{\theta}_k^{\mathrm{dp}} - \theta\right)$$

given the presence of variance heterogeneity in the estimates $\widehat{\theta}_k^{\mathrm{dp}}$, coupled with their interdependence, our approach necessitates turning to the central limit theorem as posited by Peng (2010) within the framework of nonlinear expectations.

We assume that $\mathbb{E}\left(\widehat{\theta}_k^{\mathrm{dp}}\right) = \mathbb{E}\left(\widehat{\theta}_k\right) = \theta$ or $= \theta + O_p\left(\frac{1}{\sqrt{r_k}}\right), \mathrm{var}\left(\sqrt{r_k}\widehat{\theta}_k\right) = \sigma_f^2, \ \mathbb{E}\left(\left|\widehat{\theta}_k^{\mathrm{dp}}\right|^3\right) < \infty$, and denote the bag-specific variance $\lambda_k^2$ with

$$
\begin{aligned}
\bar{\lambda}^2 &= \max\left\{\mathrm{var}\left(\sqrt{r_1}\widehat{\theta}_1^{\mathrm{dp}}\right), \cdots, \mathrm{var}\left(\sqrt{r_k}\widehat{\theta}_k^{\mathrm{dp}}\right)\right\} \\
&= \sigma_f^2 + \bar{\gamma}^2 \\
\underline{\lambda}^2 &= \min\left\{\mathrm{var}\left(\sqrt{r_1}\widehat{\theta}_1^{\mathrm{dp}}\right), \cdots, \mathrm{var}\left(\sqrt{r_k}\widehat{\theta}_k^{\mathrm{dp}}\right)\right\} \\
&= \sigma_f^2 + \underline{\gamma}^2
\end{aligned}
$$

It can be verified that $S_K(\theta)$ obeys G-Normal distribution Peng (2010) which is defined in Definition 3.

**Definition 3** (**G - Normal distribution**). *Let $\mathcal{P}_Y$ be a set of probability measures defined on the space $(\Omega, \mathcal{F})$. A measurable function $Y : \Omega \mapsto \mathbb{R}$ is said to follow a Normal distribution with lower variance $\underline{\lambda}^2$ and upper variance $\bar{\lambda}^2(0 \leq \underline{\lambda} \leq \bar{\lambda})$, if, for every Lipschitz function $\varphi$,*

$$\sup_{P \in \mathcal{P}_Y} \mathrm{E}_P[\varphi(Y)] = \sup_{P \in \mathcal{P}_Y} \int_\Omega \varphi(Y)dP = u(1, 0; \varphi)$$

where $\{u(t, y; \varphi) : (t, y) \in [0, \infty) \times \mathbb{R}\}$ is the unique viscosity solution to the Cauchy Problem

$$u_t = \frac{1}{2}\left(\bar{\lambda}^2\left(u_{yy}\right)^+ - \underline{\lambda}^2\left(u_{yy}\right)^-\right), u(0, y) = \varphi(y)$$

In the above expression $u_t = \frac{\partial u}{\partial t}, u_{yy} = \partial^2 u/\partial y^2$ and the superscripts $+$ and $-$ denote the positive and negative parts respectively and the sublinear partial differential equation is referred to a G-heat equation.

It should be emphasized that G-Normal distribution is unable to inherit the sense of traditional distribution, because G-Normal distribution of one random variable actually has distributional uncertainty and $\mathcal{P}_Y$ in this theorem denotes the collection of all possible probability measures of random variable $Y$, which is caused by the heterogeneous standard deviations from $[\underline{\lambda}, \bar{\lambda}]$ $u(1, 0; \varphi)$ represents the sublinear expectation of $\varphi(Y)$. This definition also implies when $\underline{\lambda} = \bar{\lambda} = \lambda$ the G-Normal distribution reduces to normal distribution $N\left[0, \lambda^2\right]$ and $G-$ heat equation reduces to the heat equation with

$$u(t, y; \varphi) = \int_{-\infty}^{\infty} \phi(z)\varphi(y + \kappa\sqrt{t}z)dz$$

where $\phi(y)$ is the density function of standard normal distribution.

In the following, we use the central limit theorem under nonlinear expectation theory to study the asymptotic sampling distribution of the statistic.

Let $\{\lambda_k\}_{k=1}^K \in \Sigma(\underline{\lambda}, \bar{\lambda})$ be the collection of sequences that bounded to $[\underline{\lambda}, \bar{\lambda}]$, where $\underline{\lambda}, \bar{\lambda}$ are given constants $(0 \le \underline{\lambda} \le \bar{\lambda} < \infty)$.

**Theorem 3** *For given constents $\{\lambda_k\}_{k=1}^K \in \Sigma(\underline{\lambda}, \bar{\lambda})$ and $\left\{\widehat{\theta}_k^{\mathrm{dp}}\right\}_{k=1}^K$, and for any Lipschitz function $\varphi$,*

$$u(1, 0; \varphi)$$

$$= \lim_{K \to \infty} \sup_{\{\lambda_k\} \in \Sigma(\lambda, \bar{\lambda})} \mathbb{E}\left[\varphi\left(\sum_{k=1}^K \frac{\sqrt{r_k}}{\sqrt{K}}\left\{\widehat{\theta}_k^{\mathrm{dp}} - \theta\right\}\right)\right]$$

*where $u(1, 0; \varphi)$ is given in Definition 3.*

Essentially, this theorem can be regarded as an immediate corollary of Peng's original central limit theorem Peng (2019) in terms of the language of classical probability. We can refer to the similar theoretical conclusion in the studies of Rokhlin (2015) and Fang et al. (2019), which can be seen as a generalization of the classical central limit theorem to controlled stochastic processes Peng (2019).

The theorem can be further generalized to non-identically distributed sequence $\left\{\widehat{\theta}_k^{\mathrm{dp}}\right\}_{k=1}^{\infty}$. Fang et al. (2019) introduced the detailed proof procedures and the convergence rate of Theorem 2. The function $\varphi$ can also be any Borel measurable indicator function Peng & Zhou (2020).

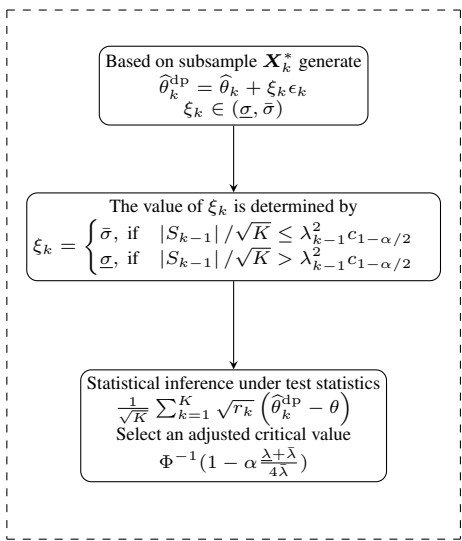

Figure 4: Flowchart of estimation and statistical inference.

