# OpenReview forum: "Nonlinear Inference Learning for Differentially Private Massive Data"
_ICLR.cc/2024/Conference — Submitted to ICLR 2024_

### Official Review · Reviewer_AGza · 2023-10-17

**Soundness:** 2 fair
**Presentation:** 2 fair
**Contribution:** 1 poor
**Rating:** 3
**Confidence:** 4

**Summary:**

This paper proposes a differentially private algorithm for computing Bag-of-Little-Bootstrap estimators of unknown population parameters. The paper further derives the algorithm’s asymptotic properties and studies the algorithm’s numerical performance.

**Strengths:**

The mathematical notation is easily to follow. The algorithm and its associated inference methods are clearly described.

**Weaknesses:**

1. *Weak motivation*. The motivation for the proposed algorithm is not convincing. It is claimed, in the last paragraph of Section 2.2, that there are two reasons for considering the paper's new algorithm: (1) dependence across bootstrap estimates, and (2) heteroskedasticity of Gaussian noise terms due to unequal sensitivities of the bags. It is unclear whether the proposed algorithm addresses either issue. The bootstrap estimates continue to be dependent in the proposed algorithm: the dependence is inherent to the bootstrap method/sampling with replacement, whether or not differential privacy is required. In fact, the very reason why Peng's central limit theorem is needed in the asymptotic analysis is the dependence across bootstrap estimates. For hetereoskedasticity, the additive noise terms in the proposed method remain heteroskedastic, just to a lesser degree: their variances are now one of the two values defined at the bottom of page 4, rather than B different values.

2. *Questionable utility*. On the point of turning B different noise variances into two different values, it raises further concerns about the proposed method’s utility. As the paper observes below Theorem 1, the resultant noise variances are guaranteed to be greater than their counterparts in the naive method. Then, it seems that the only gain from using the more sophisticated method proposed in this paper, is that there is a clean expression of asymptotic distribution. After all, the amount of noise added in the proposed method is greater than those in the naive method, and there is unlikely to be any improvement in statistical accuracy (unless the authors are able to prove that there is an improvement, either theoretically or numerically). Without justifying the benefit of using a more sophisticated method, the paper would read like studying a sophisticated method for the sake of studying it.

**Questions:**

Is the naive method more accurate than the proposed algorithm? Given $K$ non-DP estimates from the non-overlapping bags, $\hat\theta_1, \hat\theta_2, \ldots, \hat\theta_K$, the additive noise terms $u_1, u_2, \ldots, u_K$ needed for DP are independent Gaussian RVs, and therefore the aggregate estimator $\hat\theta_{DP} = K^{-1}\sum_{k=1}^K (\hat\theta_k + u_k)$ can be easily decomposed as the sum of two terms, a non-DP average and a Gaussian term $K^{-1}\sum_{k=1}^K u_k$. Considering Item 2 in "Weaknesses", the resultant utility of naive method will likely be easy to show, and better than that of the proposed algorithm. What is the reason for using the paper's new method then?

---

### Official Review · Reviewer_YJor · 2023-10-18

**Soundness:** 2 fair
**Presentation:** 1 poor
**Contribution:** 1 poor
**Rating:** 1
**Confidence:** 3

**Summary:**

The paper proposes a DP statistical inference method that is based on the bag of little bootstraps. The method splits the data into several subsets, estimates the parameter of interest on each subset, adds noise for DP, and finally combines the estimates. The statistical properties of the combined estimator are analysed with nonlinear expectation theory, which allows computing the usual uncertainty estimates of interest like confidence intervals and hypothesis tests. The method is evaluated on a toy experiment estimating the mean of a univariate Gaussian.

**Strengths:**

I don't think anyone has applied nonlinear expectation theory to DP statistical inference, but it's difficult to find other strengths in the paper.

**Weaknesses:**

The paper is very difficult to understand, as it relies on the theory of nonlinear expectations that is not properly explained in the paper, and is used as if the readers were very familiar with the theory. As a result, it is not possible to tell why the $\hat{\theta}^{dp}_k$ estimator is defined the way it is, or whether this definition makes any sense. The explanation in Sections 2.3 and 3 need to be dramatically expanded, and the relevant parts of the underlying theory need to be explicitly introduced to account for the audience at an ML conference. Some of the notation is also not defined:
- $F$ in Section 2.1
- $n_k$, $\mathcal{F}_k$, $\Sigma$ in Section 2.3
- $S_0$ and $\lambda_0$ in Section 2.3, needed to calculate $\xi_1$
- $\overline{\lambda}$ and $\underline{\lambda}$ in Section 3

The sensitivity of $\hat{\theta}_k$ in the experiment is not correct. The datapoints in the experiment are sampled from a Gaussian, so they could be any real numbers, making the sensitivity infinite. Getting a finite sensitivity would be possible by clipping the datapoints, but that is not mentioned in the paper. Definition 1 also has this problem, as it is possible for the sensitivity to be infinite.

The paper does not make a clear connection to large-scale datasets, despite advertising it in the abstract and introduction. The related work that is discussed in the introduction is also mostly related to large-scale data analysis, and overlooks work on DP statistical inference. Two especially relevant works are Covington et al. (2021) and Evans et al. (2023), which propose similar DP inference methods that also use the bag of little bootstraps. These seem to solve the same problem the paper's method does, and should be compared with experimentally.

References:
- Covington et al. "Unbiased Statistical Estimation and Valid Confidence Intervals Under Differential Privacy" arXiv 2021
- Evans et al. "Statistically Valid Inferences from Privacy-Protected Data" American Political Science Review 2023

Minor comments:
- "Gaussian differential privacy" refers to another definition of DP, not $(\epsilon, \delta)$-DP with the Gaussian mechanism. Replacing the phrase with "Gaussian mechanism" or something similar would be less confusing.
- $\delta = 0.01$ is very large compared to commonly used $\delta$ values. $\delta = 1 / n$ for $n$ datapoints allows mechanisms that are clearly non-private, so it is recommended to use $\delta \ll 1 / n$. Also, $\epsilon$ is typically the value that is varied to investigate different privacy strengths, not $\delta$.
- Font size in all figures is too small.
- Figures 4 and 5 referenced in Section 4 are missing.

**Questions:**

- What exactly is the confidence band on the y-axis in Figure 3?

---

### Official Review · Reviewer_f45X · 2023-10-28

**Soundness:** 3 good
**Presentation:** 3 good
**Contribution:** 3 good
**Rating:** 3
**Confidence:** 5

**Summary:**

The paper discusses using the Differential Privacy (DP) mechanism that integrates with the Bag of Little Bootstraps (BLB) method to enable comprehensive statistical analysis while safeguarding individual privacy. The proposed mechanism addresses variability and uncertainty in noise variance. It estimates distributions using the central limit theorem and uses nonlinear expectation theory to analyze the asymptotic distribution of statistics, allowing for hypothesis testing and parameter estimation. The primary contributions of this paper include pioneering differential privacy in static massive data using a novel algorithm, providing a general and scalable differentially private algorithm adaptable to various methods, analyzing the asymptotic distribution of statistics using nonlinear expectation theory, constructing parameter statistics, and hypothesis testing. The algorithm preserves data privacy through average-based aggregation and random noise while estimating sample statistics. The proposed mechanism effectively meets the privacy preservation requirements without compromising statistical inference, as validated by data simulation studies.

**Strengths:**

1) The paper presents a new mechanism for Differential Privacy (DP) that integrates with the Bag of Little Bootstraps (BLB) method for statistical inference on large-scale datasets while preserving data privacy.

2) The paper uses nonlinear expectation theory and the central limit theorem to address noise variance variability and estimate distribution uncertainty under differential privacy.

**Weaknesses:**

1) This paper lacks a detailed analysis of the proposed mechanism's computational complexity, which may limit its scalability to large datasets.

2) The paper does not consider addressing the impact of the proposed privacy-preserving mechanism on the utility of the data. In other words, it needs to evaluate how much information is lost because of the mechanism. Further studies are required to assess the trade-off between privacy and utility in the proposed mechanism.

**Questions:**

What are the real-world problems you will solve by combining these DP and BLB in static massive data?

---

### Official Review · Reviewer_YsDk · 2023-10-31

**Soundness:** 2 fair
**Presentation:** 2 fair
**Contribution:** 2 fair
**Rating:** 3
**Confidence:** 4

**Summary:**

This paper introduces an algorithm to provide differentially private estimates of sample statistics using the bag of little bootstraps. Asymptotic properties of the resulting estimator are given and limited experiments are presented to demonstrate the validity of the theory in a simple setting.

**Strengths:**

The paper tackles an interesting though not entirely novel problem of releasing DP statistics from big-data that yield valid statistical inferences. The paper makes an effort to ensure the statistical properties of the algorithm are well understood which is important for applied researchers looking to do inference with DP.

**Weaknesses:**

1. The contributions of the paper are overstated. 'Pioneering differential privacy in static massive data' is not a fair claim; this is a subject that has received considerable prior attention in the literature. Moreover, the use of BLB in DP algorithms is not itself new, see both 'Statistically Valid Inferences from Privacy Protected Data' Evans et al. (2023) and 'Unbiased Statistical Estimation and Valid Confidence Intervals Under Differential Privacy' Covington et al. (working paper). Both these papers use BLB within the sample-and-aggregate framework thus bypassing some of the theoretical problems this paper addresses, it is not clear why this approach was not taken in this paper?

2. More generally, the paper is lacking many relevant citations on statistical inference with DP.

3. The introduction needs to be re-written to convey the key ideas behind the methodology and how it contrasts to existing solutions.

4. Algorithm 1 seems incomplete, in particular it would be useful to map the computation steps to formulas, currently it would not be possible to implement this approach by looking at Algorithm 1 alone.

5. The simulation study is also very limited, showing just a small illustrative example of the method in a very simple setting. The evaluation metrics are also weak, why is the coverage and the validity of the variance estimate not shown? It also seems important to demonstrate the usefulness of the method for more sophisticated estimands since gaussian mean estimation with DP is already a well studies problem in the DP literature. If on the contrary the paper wants to claim this is a better method, than comparisons to existing algorithms should be given.

**Questions:**

How do you deal with possible bias introduced by clipping if the sensitive cannot be bounded apriori?

What are the sample sizes in the experiments? The paper claims this is a method for 'big data', it would help to do more to substantiate the importance of this type of algorithm in this setting (e.g. by showing existing approaches do not work).

Relatedly, why were comparisons to existing algorithms to solve this problem not presented?

---

### Meta-Review · Area_Chair_tSQQ · 2023-12-04

**Metareview:**

The paper proposes a method for DP statistical analysis on large data using bag of little bootstraps.

Strengths: careful statistical analysis of results under a novel application of DP.

Weaknesses: all reviewers recommend rejection for many reasons, including lack of comparison with relevant related work, unaccessible presentation for the ICLR audience, questions about methodology and weak experimental evaluation.

From my own quick reading of the paper, I am further concerned of what appears to be misuse of the term "differential privacy" on the level of subsamples rather than the original data set. For any future submissions, I would highly encourage the authors to adopt a different term or add a qualifier to make it clear that this is in general not equivalent to the standard definition of DP on the level of the full data set.

**Justification For Why Not Higher Score:**

All reviewers recommend rejection.

**Justification For Why Not Lower Score:**

N/A

---

### Decision · Program_Chairs · 2024-01-16

Reject